# Measuring luteinising hormone pulsatility with a robotic aptamer-enabled electrochemical reader

Shaolin Liang[1,2,3], Andrew B. Kinghorn[1], Margaritis Voliotis [4], Julia K. Prague[2], Johannes D. Veldhuis[5], Krasimira Tsaneva-Atanasova [4], Craig A. McArdle[6], Raymond H.W. Li[7], Anthony E.G. Cass [3], Waljit S. Dhillo[2] & Julian A. Tanner [1]

Normal reproductive functioning is critically dependent on pulsatile secretion of luteinising hormone (LH). Assessment of LH pulsatility is important for the clinical diagnosis of reproductive disorders, but current methods are hampered by frequent blood sampling coupled to expensive serial immunochemical analysis. Here, we report the development and application of a Robotic APTamer-enabled Electrochemical Reader (RAPTER) electrochemical analysis system to determine LH pulsatility. Through selective evolution of ligands by exponential enrichment (SELEX), we identify DNA aptamers that bind specifically to LH and not to related hormones. The aptamers are integrated into electrochemical aptamer-based (E-AB) sensors on a robotic platform. E-AB enables rapid, sensitive and repeatable determination of LH concentration profiles. Bayesian Spectrum Analysis is applied to determine LH pulsatility in three distinct patient cohorts. This technology has the potential to transform the clinical care of patients with reproductive disorders and could be developed to allow real-time in vivo hormone monitoring.

[1] School of Biomedical Sciences, LKS Faculty of Medicine, The University of Hong Kong, Hong Kong, China. [2] Section of Endocrinology and Investigative Medicine, Imperial College London, London, SW7 2AZ, UK. [3] Department of Chemistry, Imperial College London, London, SW7 2AZ, UK. [4] Department of Mathematics and Living Systems Institute, College of Engineering, Mathematics, and Physical Sciences, University of Exeter, Exeter, EX4 4QD, UK. [5] Endocrine Research Unit, Mayo School of Graduate Medical Education, Mayo Clinic, Rochester, MN 55905, MN, USA. [6] Laboratories for Integrative Neuroscience and Endocrinology, Bristol Medical School, University of Bristol, Bristol, BS1 3NY, UK. [7] Department of Obstetrics and Gynaecology, LKS Faculty of Medicine, The University of Hong Kong, Hong Kong, China. These authors contributed equally: Anthony E. Cass, Waljit S. Dhillo, Julian A. Tanner. Correspondence and requests for materials should be addressed to A.E.G.C. (email: t.cass@imperial.ac.uk) or to W.S.D. (email: w.dhillo@imperial.ac.uk) or to J.A.T. (email: jatanner@hku.hk)

Normal reproductive function is governed by a highly orchestrated pattern of hormonal feedback across the hypothalamic–pituitary–gonadal (HPG) axis. The pulsatile release of luteinising hormone (LH) is a critical element for downstream regulation of sex steroid hormone synthesis and the production of mature oocytes[1]. Altered patterns of LH pulse secretion have been linked to hypothalamic dysfunction, resulting in numerous reproductive disorders, including polycystic ovary syndrome (PCOS)[2], hypothalamic amenorrhoea[3] and delayed or precocious puberty[4].

It is not currently feasible in routine clinical practice to measure LH pulsatility to determine altered secretion patterns because to do so is extremely resource intensive. Peripheral blood sampling is necessary every 10 min for at least 8 h, and serial analysis by immunochemical assay is expensive[5,6]. Most of the data measuring LH pulsatility are therefore in animals, including rodents, sheep and monkey[7–9] or in some human studies from specialist clinical research groups[10,11]. There are three major issues currently preventing widespread clinical LH concentration profile measurement and its pulsatility analysis: (1) LH concentration profile's resolution is restricted by the sampling protocol and immunochemical assay—there is no method capable of real-time monitoring of LH pulsatility. (2) Measurement of LH concentration profile is restricted by cost when serial clinical chemiluminescent immunoassays are used (~£20 per sample, 50 samples required for one patient). (3) LH pulsatility analysis is challenging as it typically requires advanced algorithms that can appropriately and efficiently account for the inherent biological variation, pulse-by-pulse variability and physiological factors impacting on hormone secretion and calculation, including elimination[12]. There is a clear unmet medical need for better translational technologies that could enable routine clinical LH pulsatility analysis for patients with reproductive disorders.

Better technologies for hormone sensing, that could in the future be adapted for continuous sensing as an individual undergoes their normal daily routine, would revolutionise the clinical care of patients with reproductive disorders. Glucose monitoring has transformed diabetes care[13], and similarly, intrinsic chemical reactivities of lactate[14] and neurotransmitters[15–17] such as dopamine, serotonin, glutamate and acetylcholine using specific enzymes for detection have had significant impact. However, such detection approaches are not generalisable for larger peptides, hormones and proteins such as LH, as they all depend on specific enzymes recognising particular small-molecule metabolites[18]. Antibodies have been the mainstay for quantification of larger molecules in molecular diagnostics for decades, yet have serious challenges for repeated sensing operations (not easy to be regenerated), and so are not suitable for continuous sensing applications necessary for hormone monitoring. The emerging technology of electrochemical aptamer-based (E-AB) sensing could potentially offer a generalisable approach for LH pulsatility measurement.

E-AB sensors take advantage of the intrinsic properties of nucleic acid aptamers to specifically bind to a molecular target and undergo a reversible conformational change. The conformational change can be measured through the electrochemical signal response, governed by distance between an electrochemical reporter covalently linked to the aptamer and a gold sensor surface. Binding events can be indicated by a change in the electrochemical signal either in a current or an impedance format[19]. E-AB sensors are also relatively straightforward to prototype from development to a device[20]. Continuous sensing of drug pharmacokinetics using an E-AB sensor has been demonstrated in living animals[21,22], although not yet in humans. The coupling of easily modified aptamers together with rapid prototyping technologies enables E-AB sensors to be developed for different applications. E-AB sensors neither require washing, separations, chromatographic steps, nor expensive equipment, so are excellent choices as an approach for continuous, low-cost sensing that in future could be adaptable for in vivo sensing of numerous hormones.

To this end, we develop an LH-specific aptamer and engineered it for specific electrochemical sensing in a robotic platform. Using this robotic platform, we analyse patient samples from three different patient cohorts with distinct LH pulsatility profiles to determine LH concentration and then used Bayesian spectrum analysis (BSA) to analyse LH pulsatility[23,24]. This approach demonstrates promise for enabling more widespread LH pulsatility analysis in clinical practice, and lays a foundation for future in vivo E-AB-based hormone monitoring. This technology could be applied using a similar approach to the in vivo continuous measurement of other hormones that would revolutionise the diagnosis and treatment of patients with endocrine disorders.

## Results

**The aptamer-enabled electrochemical robotic platform.** In order to address the challenges of clinical LH monitoring, we designed an aptamer-enabled electrochemical robotic platform based on the integration of multiple technologies. The platform consists of a potentiostat, a laptop interface, an open-source programmable robotic platform and a customised 96-well plate electrochemical sensing system embedded with an LH aptamer sensor (Fig. 1a). We designed a 96-well-plate assay system that allows the automation of continuous testing of patient samples under the open-source robotic platform with a rapid measurement interval time of 50 s for each sample. The key component of this assay system is the wire electrode functionalised with an LH aptamer labelled with an electrochemical reporter. The quantification of the LH level is achieved by analysing the electron transfer kinetic changes upon binding of LH via a three- microwire electrode system held with a 3D-printed part (Fig. 1a). Only 50 µl of sample is required for the measurement, and LH concentration profile can be generated from the signal response of the potentiostat. The pulse interval information can then be extracted using a BSA method to enable identification of LH pulsatility. Three individual patients LH concentration profiles are shown in Fig. 1c (red, menopausal woman—high LH levels with pulsatility; blue, healthy female—normal LH pulsatility; grey, female patient with hypothalamic amenorrhoea (HA) patient—reduced LH levels and pulsatility). The BSA method identified their pulse interval ranges.

**Developing a high-affinity LH-specific aptamer.** The LH aptamer was generated via a nitrocellulose-membrane-based SELEX method (Fig. 2a). In order to obtain a highly specific LH aptamer, we applied counter selection against follicle-stimulating hormone (FSH), a glycoprotein sharing a structural similarity with LH. Both LH and FSH share the same α-chain and hence it is necessary to evolve an aptamer that can differentiate between the two through interactions with the β-chain or at the interface between the two subunits. The gel electrophoresis image after each round of selection was used to monitor the selection process. Twenty rounds of selection were performed before sequencing. We obtained eight major sequences. B10, B11 and B35 only showed one base variation between them. Enzyme-linked oligonucleotide assay (ELONA) was used to screen the DNA aptamers for their affinity in binding to LH (Supplementary Fig. 1). The concentration–response curve obtained by ELONA of the best LH-binder aptamer B23 is shown in Fig. 2b. No significant FSH interaction was observed. The apparent equilibrium dissociation constant ($K_D$) was estimated to be 321 nM. Further cross-reaction

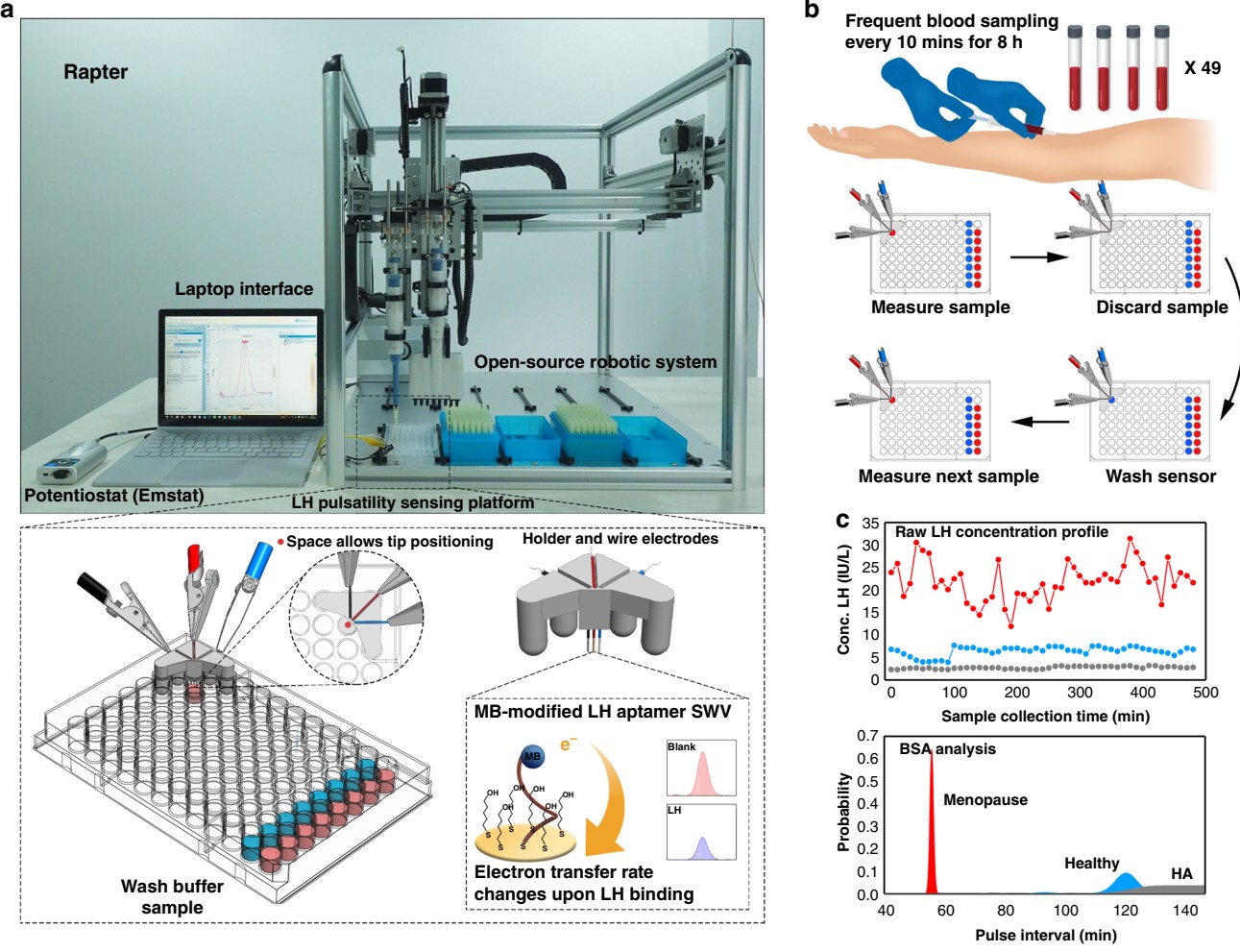

**Fig. 1** Overview of the Robotic APTamer-enabled electrochemical reader (RAPTER). **a** RAPTER design. RAPTER consists of a potentiostat, laptop interface, open-source liquid-handling robot and the 96-well-plate sensing platform. The quantification of the LH level is achieved by analysing the electron transfer kinetic changes (interrogated via square-wave voltammetry (SWV)) upon binding of LH under a three microwire electrode system held within a 3D-printed holder. **b** A series of samples can be loaded onto the 96-well platform, and the measurement process can be automated by the programmable liquid-handling robot. Serum samples collected from frequent blood sampling analysed by the immunochemical test (antibody test) were used as a comparison with test RAPTER performance in measuring LH concentration in human clinical samples. **c** The time-course data (LH concentrations at 10-min intervals over 8 h) can be further analysed by BSA to identify pulse intervals (and their likelihood) and to classify patients. Source data are provided as a Source Data file

studies were conducted to investigate the specificity of B23 aptamer against multiple proteins using ELONA and electrophoretic mobility shift assay (EMSA) (Fig. 2c). No significant nonspecific interactions were observed in either assay. A clear concentration–response relation was observed in the EMSA gel. We also performed DNA–protein interaction study using surface plasmon resonance (SPR) to investigate the binding kinetics of LH and B23 aptamer. SPR response data from a titration study were fitted to a 1:1 binding model (Supplementary Fig. 2). The estimated $K_{off}$ was $1.2 \times 10^{-3}\,s^{-1}$. It is a relatively fast off-rate (slow off-rates are normally $< 10^{-5}\,s^{-1}$). This feature benefits a continuous sensing-format design by allowing fast regeneration. A circular dichroism (CD) study of B23 investigated the conformational change dynamics upon binding of LH. Figure 2d shows the CD spectra before and after adding $1\,\mu M$ LH in buffer. The negative peak at 240 nm becomes more negative and the positive peak at 280 nm increases after binding, providing evidence of a structure-switching conformational change of the aptamer upon LH binding. This feature is important as in the electrochemical biosensor design, conformational change is needed to generate the signal change on target binding.

**A wire electrode system for LH sensing**. We applied the Methylene Blue (MB)-modified B23 aptamer onto a wire-based three-electrode system. This system consists of wires (diameters ranging from 0.5 to 1 mm) with different materials (working: Au, reference: Ag/AgCl, counter: Pt). All of the electrodes were wrapped by PVC insulation tape with 5 mm left at the detection end and the soldering end, and epoxy was used to seal the connection parts to avoid water leakage. All of the electrodes were assembled into a 3D-printed holder that fits a single well of a 96-well plate whilst avoiding electrical shorting, allowing enough space for tip positioning (Fig. 3a). Target binding induces a conformational change in the LH aptamer that alters the rate of electron transfer between the MB and the electrode. This change is expected to produce a readily measurable variation in current at the MB reduction peak when the sensor is interrogated using square-wave voltammetry (SWV). The relative change in the current therefore provides a direct measurement of LH concentration. The cleaning for the electrode was achieved by repeat cyclic voltammetry (CV) scanning in $0.5\,M\ H_2SO_4$ to remove impurity layers (Fig. 3b). Since the relative change of this current is highly dependent on SWV frequency, we interrogated the

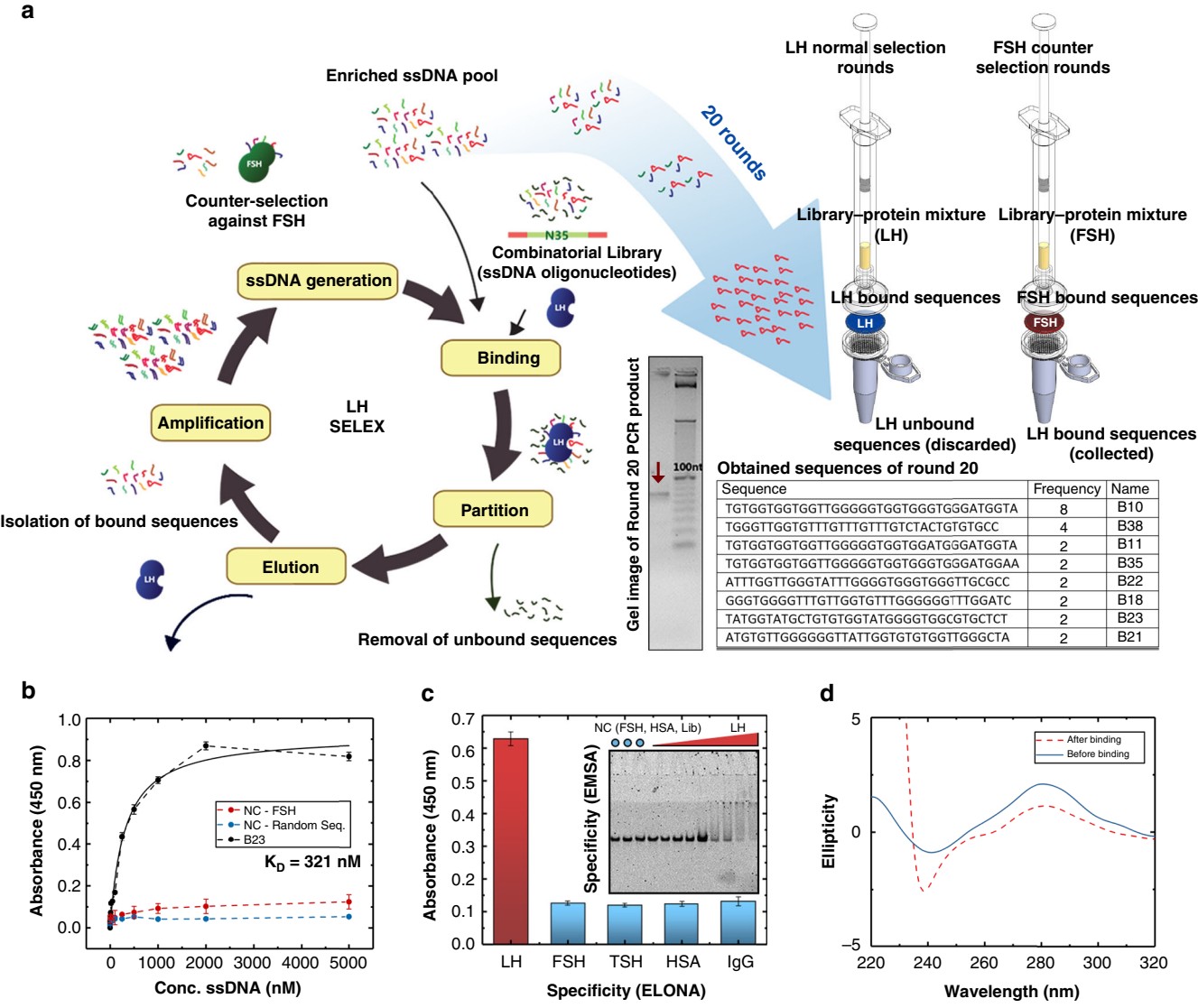

**Fig. 2** Characterisation of the LH aptamer. **a** Nitrocellulose membrane-based SELEX method for LH aptamer generation. The LH aptamer selection was performed by 20 rounds of binding, partitioning, eluting, amplification and ssDNA generation steps. Nitrocellulose membrane filtration was used to partition the LH-bound sequence in the LH selection rounds and FSH-bound sequences in the FSH counter selection round. Thirty-eight clones of the last-round library were sequenced. **b** Affinity study of the B23 aptamer. The concentration–response curve of B23 is shown in black. Two blue lines indicate the negative controls, one is the response of truncated B23 against FSH (red) and the other is the response of a random sequence against LH (blue). Means ± s. d. from three independent experiments are shown ($n = 3$). **c** Specificity study of B23 via ELONA and EMSA. The red column represents the colourimetric response of adding 1 µM B23 onto the LH plate and the blue columns represent the response of adding 5 µM of B23 to other nonspecific proteins. The gel image shows the specificity study of B23 using EMSA. The blue dots represent the negative controls of the experiment with 1 µM of B23 and 5 µM FSH, 1 µM of B23 and 5 µM HSA and a random sequence with 5 µM LH. The rest of the lanes show 1 µM of B23 with a series of LH concentration ranging from 0 to 5000 nM. Means ± s.d. from three independent experiments are shown ($n = 3$). **c, d** Circular dichroism (CD) of B23 before and after adding 5 µM of LH. The red dashed line represents the change after binding, a negative peak at 240 nm increased and a positive peak at 280 nm decreased. Source data are provided as a Source Data file

sensor at multiple SWV frequencies. We observed a signal off response in the majority of frequencies we tested (from 10 to 500 Hz), and a signal on response below 10 Hz (Fig. 3c). As it requires more than 50 s for the potentiostat to perform SWV scan of frequencies below 10 Hz, we decided to use 100 Hz in the signal off range as the optimal frequency for LH measurement. The detailed electrochemical characterisation of the electron transfer rate can be found in the Supplementary materials (Supplementary Fig. 3). We investigated the sensor's performance for LH detection using the SWV parameters optimised for maximising the gain. The signal change upon adding spiked LH in PBS buffer was fitted to a Langmuir isotherm and a $K_D$ of 259 nM was obtained.

The sensor achieved a limit of detection (LOD) in PBS buffer of 10.9 nM, with a dynamic range of 5–500 nM. Furthermore, a high concentration of FSH (1 µM) did not affect the sensor (Fig. 3d, e).

**Robustness of RAPTER in undiluted clinical serum samples.** We further engineered the wire system to a robotic platform for the assay automation. Electrodes were assembled into a custom-designed 3D-printed electrode holder to avoid electrical shorting and contact with the pipette tips while performing the measurement and wash steps (Fig. 1b). The liquid-handling robot Opentron is an affordable open-source platform that allows a

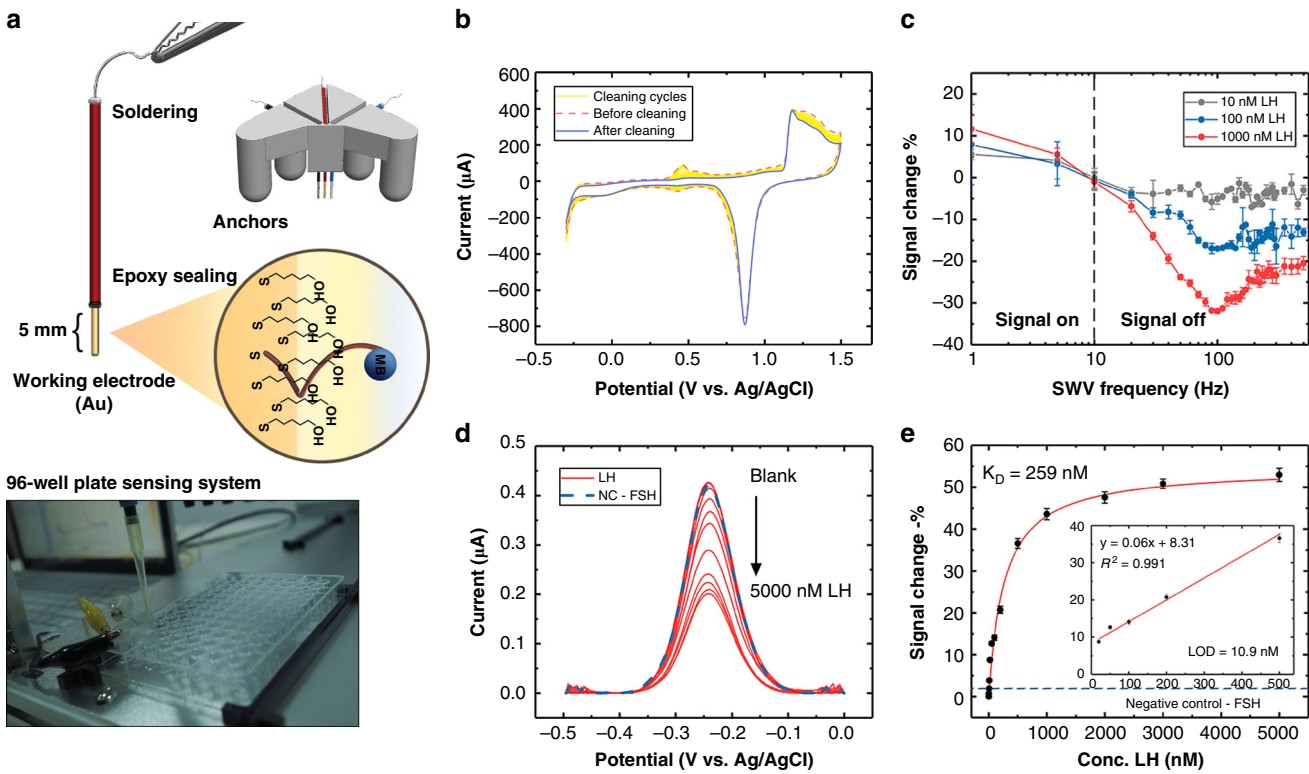

**Fig. 3** Characterisation of the wire electrochemical system. **a** Illustration of the wire electrochemical system. Gold, Ag/AgCl and platinum wires were used as the working, reference and counter electrode, respectively. All the wires were insulated by PVC insulation tape and left two exposed ends for sensing and connecting. All the wires were placed into a 3D-printed holder with anchors that can sit on a single well of a 96-well plate. **b** Electrochemical cleaning of the wire was performed using CV by repeat scanning from −0.3 to 1.5 V in 0.5 M sulphuric acid until typical clean gold cyclic voltammograms were observed. **c** Signal change at different SWV frequencies. MB peak signals of the LH aptamer were recorded under scans in different SWV frequency settings. We applied three concentrations (10, 100 and 1000 nM) of LH to investigate the signal change. Means ± s.d. from three independent experiments are shown ($n = 3$). **d** Square-wave voltammogram of wire-based LH aptamer sensor. All the curves were net currents obtained by performing SWV scan from −0.5 to 0 V, with 100-Hz frequency and 25-mV amplitude. Eight concentrations of LH were used: 0, 10, 50, 100, 200, 500, 1000 and 2000 nM. Error bars represent standard deviation ($n = 3$ replicates). **e** Concentration–response curve. The % of the signal change indicates that the change is a signal off model. The $K_D$ was estimated to be 259 nM and the dynamic range was from 10 to 500 nM. Means ± s.d. from three independent experiments are shown ($n = 3$). Source data are provided as a Source Data file

pre-programmed protocol to meet individual experimental requirements. Simple Python commands control automation of the assay steps (Fig. 1b, right). We investigated the signal response and sensor regeneration performance of RAPTER using serum samples from patients diagnosed with hypothalamic amenorrhoea (who have no LH/suppressed LH secretion). We performed both CV and SWV on the undiluted serum samples. In the cyclic voltammogram shown in Fig. 4a, we observed a potential shift from −0.24 to −0.28 V, and the $H_2O$ oxidation became weaker (the negative tail of the blue curve became shorter in the red curve). This indicates adsorption of proteins in the serum onto the electrode surface. However, in the square-wave voltammogram presented in Fig. 4b, there was no obvious signal reduction, but a significant peak shift corresponding to the CV study from −0.24 to −0.28 V. This indicates that nonspecific absorption does not change the aptamer structure. Regeneration performance was also evaluated by applying the maximum concentration (500 nM) of added LH to a serum sample from a patient with hypothalamic amenorrhoea (these patients have very low LH levels). The signal can be regenerated to more than 93% at concentrations under 500 nM. However, when we applied the LH concentration at saturation levels (5000 nM), we failed to recover the sensor (Fig. 4c). The concentration–response curve and LOD were also investigated using an added LH clinical sample. As can be seen in Fig. 4d, a $K_D$ of 358 nM was obtained,

with a similar LOD (10.7 nM) and dynamic range (1–500 nM) achieved with added LH in PBS buffer. This is within the LH range in human blood sample excluding the LH-surge period (we converted our result in nM into IU/L according to 96/602 standard reference, 1 IU/L equals ~5.6 nM). We also performed a pilot study to ensure RAPTER performance in six human samples with known different LH values (ranging from 3.7 to 29.9 IU/L). The results detected by RAPTER are comparable with the results determined by immunochemical assay (Fig. 4e).

**Comparison of RAPTER and clinical immunochemical assay.** Having shown that RAPTER can reliably operate with clinical, undiluted patient samples, a larger-scale evaluation was performed using 441 serum samples from three patient cohorts known to have different LH concentrations and pulsatility profiles as a proof-of-concept experiment (each participant had 49 samples, measured every 10 min during an 8-h study). The three patient cohorts were young fertile females with normal menstrual cycles (normal LH pulsatility and LH range 2–8 IU/L), menopausal women (disordered LH pulsatility with high LH levels range 15–70 IU/L), and women with hypothalamic amenorrhoea (suppressed or absent LH pulsatility with very low LH levels range 0–2 IU/L). Sampling was performed every 10 min for 8 h to collect a sufficient amount of blood for both RAPTER and

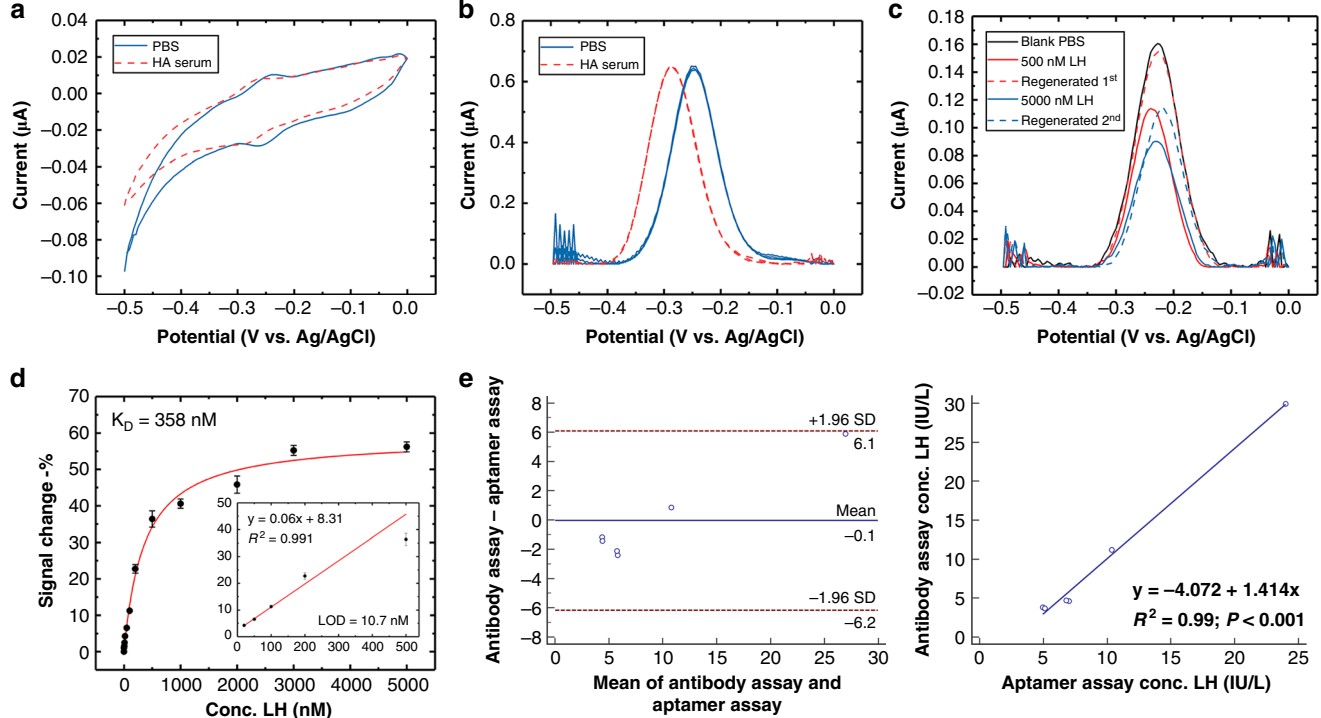

**Fig. 4** Performance of RAPTER in undiluted clinical serum samples. **a** CV scan of the LH aptamer sensor in PBS and then in undiluted hypothalamic amenorrhoea serum. **b** SWV of the LH aptamer sensor in PBS and then in undiluted hypothalamic amenorrhoea serum. **c** Regeneration under an undiluted serum condition. The dashed line represents the regenerated peak obtained after using the automatic platform washing step. **d** Concentration–response curve using SWV methods. The % of the signal change indicates that the change is a signal off model. The $K_D$ is estimated to be 358 nM, and the dynamic range is from 5 to 500 nM. Means ± s.d. from three independent experiments are shown ($n = 3$). **e** Pilot study on measuring LH concentration. Six samples with a known wide variation in the range of LH concentration measured by the aptamer and clinical gold standard immunochemical assay. Bland–Altman plot and linear regression plot of the LH concentration obtained by the two methods, $R^2 = 0.93$. Source data are provided as a Source Data file

immunochemical assay. The same patient samples were analysed using both the current gold standard clinical method for measurement of LH and FSH (immunochemical assay) and RAPTER for comparison. The RAPTER assay was performed using the setup illustrated in Fig. 5a. Two 96-well plates were used for each patient (49 samples). RAPTER runs the measurement programme with a set waiting time to synchronise the SWV measurement initiated by the potentiostat's software interface.

All obtained LH concentrations for both RAPTER and clinical immunochemical assay can be seen in Supplementary Fig. 4. To compare the methods, we performed a Bland–Altman analysis and plotted the linear regression. The RAPTER assay shows a good overall correlation with the clinical immunochemical assay ($R^2 = 0.94$) (Fig. 5c). We also demonstrated the high specificity of RAPTER for LH measurement in these clinical samples as in the menopause cohort, all patients show higher FSH values compared with LH (measured by the immunochemical assay) (Supplementary Fig. 5). We then inferred, using BSA, the effective LH pulse interval from both datasets (immunochemical assay and RAPTER assay). Figure 5d–f shows the original datasets of three representative patients from the three different patient cohorts, and Fig. 5g–i shows the corresponding results of the BSA analysis in the form of posterior probability distribution of the LH pulse interval given the data. We are able to distinguish patient cohorts based on the time interval ranges. Menopause patients had higher amplitude pulses.

## Discussion

We have presented a method for LH concentration measurement that is automated, reusable and low cost, and in future could be adapted for continuous sensing. As a proof-of-concept, we determined LH concentration in clinical patient samples with variable reproductive function, including menopausal women with high LH values, normal fertile women with normal levels of LH and women with hypothalamic amenorrhoea who had suppressed levels of LH. Our results demonstrate that RAPTER not only has comparable performance to the current gold standard clinical immunochemical system but also can compete with lower cost, lower reagent use and simpler setup (no extra reagents needed for each concentration point measurement). Furthermore, the RAPTER was able to detect the change in pulsatile LH secretion as reflected by the appropriate measurement of variable levels of LH within a patient's 8-h clinical study and with subsequent confirmation by BSA analysis. A similar distribution was achieved irrespective of whether clinical assay or RAPTER LH concentrations were used for analysis. This offers strong evidence for this platform's potential clinical utility in pulsatile hormone measurement. Also, the platform is highly adaptable for multiplexing or implanting for continuous monitoring. Multiple single-wire electrodes would be highly adaptable for multiplex hormone measurements in a single sample. A wire electrode may also be integrated with an automated blood collection system for hormone pulsatility measurement in real time[25,26]. Furthermore, E-AB in vivo continous sensing for small molecules has already been achieved in awake and ambulatory animals[27], which also shows potential for our system after further refinement of the sensing performance in whole blood.

E-AB continuous sensors have been reported to have a signal drift issue when performing the test in whole-blood samples[27]. Solutions have included better blocking reagents[28], drift correction based on electron transfer dynamics[29,30] or alternative

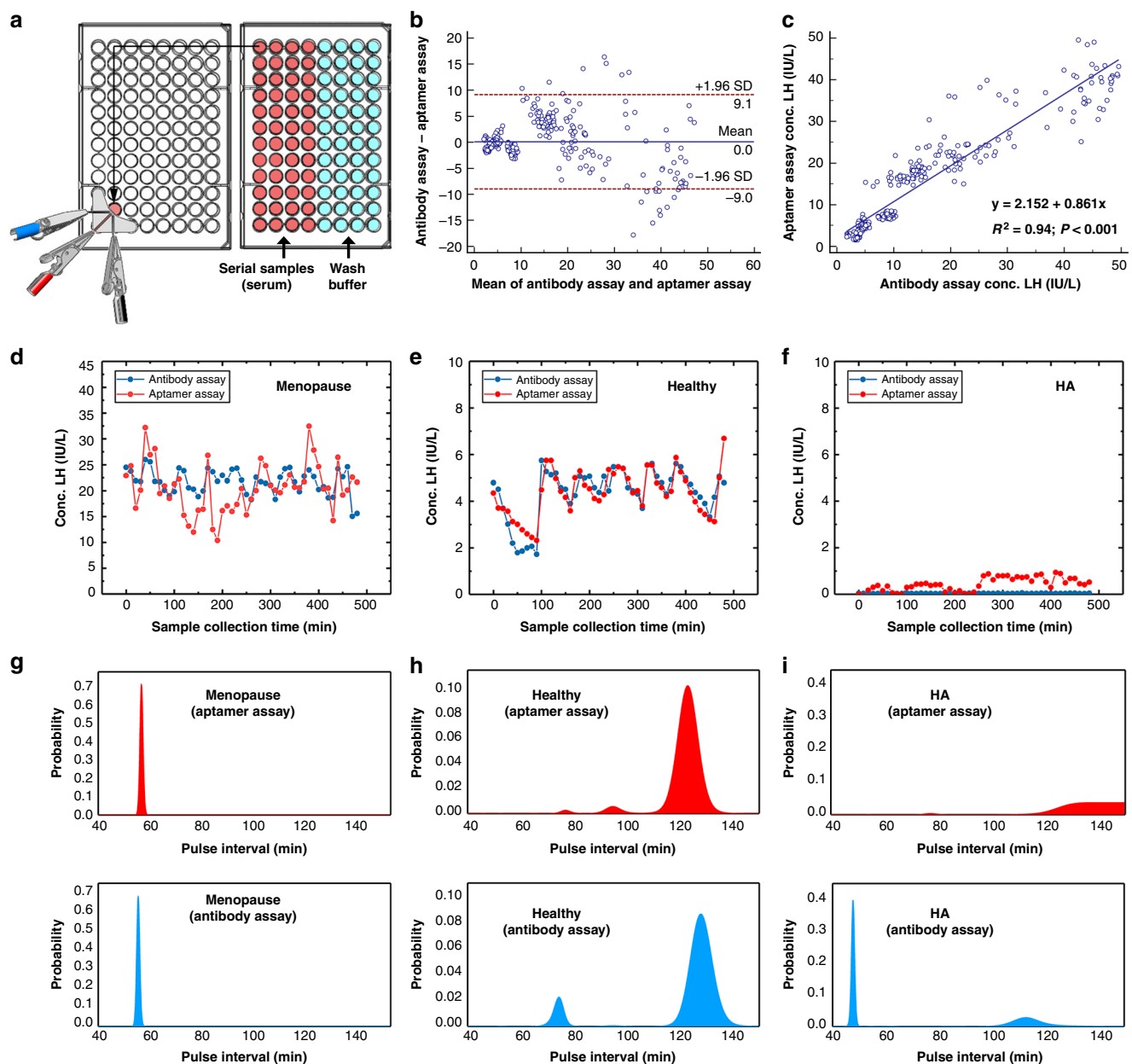

**Fig. 5** Distinguishing patient cohorts by LH pulsatility profiles using RAPTER and BSA. **a** RAPTER setting for the measurement of clinical samples. Illustration of the measurement programme design. **b** Bland–Altman analysis for the comparison of clinical assay result and aptamer assay result. **c** Linear regression of all the datasets in three patient cohorts for the comparison of RAPTER performance with clinical standard immunochemical assay. **d–f** Individual LH pulsatility profiles in women with menopause (**d**), healthy females (**e**) and hypothalamic amenorrhoea (**f**) obtained by RAPTER (red) and clinical assay (blue). **g–i** Pulse interval analysis in women with menopause (**g**), healthy females (**h**) and hypothalamic amenorrhoea (HA) (**i**) obtained by RAPTER (red) and clinical assay (blue). Source data are provided as a Source Data file

drift-free electrochemical interrogation[30]. However, these methods are highly dependent on each aptamer's intrinsic properties. Therefore, a suitable pulse analysis method is critical. BSA is well suited to analysing pulsatile hormonal dynamics, since it allows for automated drift model selection without the need to preprocess the data[23,24].

There is currently no aptamer-based LH sensing technology. Clinical assay platforms such as Roche's Cobas™, Siemens Centaur™ and Abbott's Architect™ provide antibody-based automated hormone assay options[6]. Lateral flow test strips using gold nanoparticle-labelled antibody and signal enhancers for single LH point measurements in urine for ovulation detection are also available[31]. The LH aptamer described herein offers a valuable

sensing reagent for LH detection. The biophysical and biochemical characterisation of the LH aptamer demonstrates three beneficial features: (1) high binding affinity for LH and no significant off-target binding to the related molecules; (2) a binding-dependent conformational change; (3) a relatively fast off-rate. The RAPTER utilises these features of the aptamer to perform LH pulsatility measurement. Clinical results were consistent with the specificity of the LH aptamer for LH over FSH, as the system still worked well in menopause samples where FSH levels are significantly higher than LH levels.

Taken together, the RAPTER system provides an efficient approach for LH pulsatility determination by reliably calculating varying LH concentrations within an 8-h patient study that may

be both adaptable to other hormones and for a fully continuous sensing system. Future work will focus on the applicability and generalisability of this approach for continuous hormone sensing that would revolutionise the diagnosis and treatment of patients with endocrine disorders.

## Methods

**Nitrocellulose membrane-based SELEX**. LH-specific aptamer was generated via a nitrocellulose-membrane-based SELEX method. The SELEX was started by incubation of recombinant hLH (Serna, USA) with a 75-nt long ssDNA library flanked with two primer regions. The nitrocellulose membrane was used to capture the ssDNA-LH-bound complex, and the selected ssDNA was eluted by heating the membrane in water at 95 °C for 5 min. The eluted sequences were amplified using a Pfx DNA polymerase amplification kit with a 3′-biotinylated primer. The double-stranded DNA PCR product was applied to streptavidin magnetic beads for the separation of non-biotinylated single-stranded DNAs from their complementary biotinylated strand. The beads were pre-washed three times by the 1× binding and wash (B&W) buffer recommended in the product protocol. After the pre-wash step, 95 μL of PCR product and 2 μL of 2× B&W buffer was then added to the beads and mixed for 30 min at room temperature on a rotating platform. The mixture was then applied to a magnet and the supernatant was removed. The selected single-stranded DNA sequences were separated from their complementary strands (with a biotin-modified 5′ end) by incubating with 50 μL of 0.1 M NaOH and eluting after 5 min. In total, 150 μL of binding buffer was then added and the DNA concentration was measured with a Nanodrop-2000. Counter-selection rounds against FSH were carried out after the 5th, 10th and 15th round of the selection. In these selection rounds, FSH was used instead of LH for the incubation and only the filtrate from the nitrocellulose-membrane filtration step was collected for the remaining steps. The last round of the selection Round 20, was chosen for cloning and sequence analysis. The Round 20 pool was amplified with unmodified primers and purified using the PCR clean-up system (Qiagen). The product was cloned into a PCR Blunt II TOPO vector (Invitrogen) and transformed into *E. coli* DH 5-alpha competent cells via heat shock. Forty colonies of each pool were picked and purified using the mini-prep kit (Qiagen). In total, 40 samples were prepared and sequenced (Techdragon). All the sequence data collected were aligned using Vector NTI Advanced 11.1 (Invitrogen) and the secondary structure was predicted by MFold with the solution set according to the selection buffer.

**Biochemical and biophysical characterisation of the aptamer**. ELONA was performed for four different selection rounds (Round 3, Round 7, Round 15 and Round 20). In total, 160 ng of LH and FSH in 100 μL of selection buffer were separately coated on each well of the 96-well plates overnight at 4 °C separately. After blocking with PBS Tween-20 (PBST) buffer with 1% BSA for 1 h at room temperature, the wells were washed three times with the selection buffer. A biotinylated complementary strand for the 5′-end primer region was annealed to the Round 3, Round 7, Round 15 and Round 20 pools. These biotinylated pools with different concentrations (1 μM, 500 nM and 50 nM) were added to the LH-coated wells/FSH-coated wells and incubated for 1 h at room temperature. Following three washes, streptavidin–horseradish peroxidase (HRP) conjugate was added to each well. The plate was incubated for 30 min and washed with 0.1% BSA PBST buffer. In total, 50 μL of TMB was then added to the wells and incubated for 30 min. By adding 10 μL of 1 M $H_2SO_4$, the reaction was quenched and the binding complexes were quantified by measuring the absorbance at 450 nm in a microplate reader. The further characterisation of the six aptamer candidates was performed using a similar ELONA setup but with different ranges of aptamer concentration for the titration. The last selection pool Round 20 was tested for its binding affinity to LH using an EMSA. The sample pool was prepared by magnetic bead-based ssDNA generation after the amplification of the Round 20 library. In total, 0.2 μM Round 20 ssDNA pool was then incubated with increasing concentrations of LH and FSH from 0.85 to 6.8 μM in a total volume of 10 μL at room temperature. After 1 h of incubation, the samples were resolved by double-layer non-denaturing polyacrylamide gels (6% polyacrylamide for stacking and 12% polyacrylamide for resolving) containing 1× TBE buffer. The gel was stained with SybrGold for 30 min and imaged on the GelDoc™ XR + system (Bio-Rad). The further characterisation of the B23 aptamer candidate was performed using a similar EMSA protocol, but with a different range of aptamer concentrations for the titration. In total, 1 μM of B23 was immobilised onto a plain gold chip in an overnight incubation at room temperature. After the immobilisation, 1 mL of 2 mM MSA was used to block the unoccupied binding sites on the gold surface. The plain gold chip was then placed into a flow cell of the Reichert's SPR platform. The SPR measurement was fine-tuned by adjusting the time settings and flow rates of injection, dissociation, waiting and regeneration. The final settings were: 40 min of injection, 40 min of dissociation following another 40 min of injection of the regeneration solution, 40 min of dissociation and 40 min of equilibrating at a flow rate of 2 μL/min. Different regeneration solutions were tested in order to obtain a better regeneration efficiency: 10 mM glycine at pH 2, 10 mM glycine at pH 10 and 50 mM NaOH. The same concentration of the analyte was injected six times and the relevant responses were analysed. Circular dichroism spectra of 5 μM solutions of six aptamer candidates were measured with a Jasco J-810 spectropolarimeter. In the

conformational change study, the CD spectra were obtained between 220 and 320 nm. The data gathered were the average of four scans at a scanning rate of 100 nm/min. The scan of the buffer recorded at room temperature was subtracted from the average scans for each DNA sequence. Data were processed in Excel and Origin for plotting.

**Wire LH aptamer sensor fabrication**. To fabricate the sensor, 6 cm of gold, silver and platinum wires in 1-mm diameter were first polished with P240 sandpaper gently and cleaned with absolute ethanol and ddH₂O. PVC insulation tape was used to wrap the wire electrodes for insulation. Overall, 5 mm of the metal was left exposed at both ends, and epoxy was used to seal the edges of the tape and the wire to avoid water leakage. The top end was soldered to a connector cable for the connection of the potentiostat. The silver wire was immersed in bleach overnight to form a silver chloride film. No special treatment was applied for the platinum electrode. The gold working wire electrode was further electrochemically cleaned by applying repeat scanning CV from −0.3 to 1.5 V in 0.5 M sulphuric acid until typical clean gold cyclic voltammograms were observed. Electrochemical roughening was also performed to increase the electrochemical active surface area[32]: A two-step chronoamperometry was applied to the cleaned gold wire electrode by alternating the potential between 0 and 2 V, back and forth for around 10,000 pulses in 0.5 M sulphuric acid. The cleaned, roughened gold working wire electrode was immersed into a reduced MB-modified B23 LH aptamer solution in PBS for 2 h under sealed, moist and room-temperature conditions. Subsequently, the sensor was immersed overnight in 10 mM 6-mercapto-1-hexanol (MCH) in PBS to coat the remaining gold surface. The modified gold working electrode was assembled with the platinum counter electrode and silver/silver chloride reference electrode to the 3D-printed electrode holder and stored in buffer condition before use.

**Electrochemical measurement**. All the electrochemical measurements were performed using an Emstat blue potentiostat in an electrochemical cell customised for 96-well-plate sensing with the wire LH aptamer sensor. For the SWV measurements, the sensors were interrogated from 0.0 to −0.5 V (Ag/AgCl reference electrode) with 25-mV amplitude, 4-mV steps and suitable frequency setting (ranging from 1 to 500 Hz). For the CV measurements, the sensors were interrogated from 0.0 to −0.5 V (Ag/AgCl reference electrode) at 0.1 V/s. The CV scan rate study was performed under a scanning potential ranging from −0.5 to 0 V. Seven scan rates were used: 0.2, 0.15, 0.1, 0.08, 0.05, 0.04 and 0.03 V/s. The LOD was calculated from three times standard deviation of the blank sample (PBS or HA serum without LH)/slope of the calibration line.

**Automation**. The Opentron robotic platform was calibrated before each usage. The principle is to allow the robot to remember the preset position at each slot (such as 96-well plates, tip box, trash etc.) by moving the liquid-handling arm to the preset position manually. The automated measurement cycle was programmed as the following: (1) move the sample from sample well to the detection well , (2) wait for the measurement , (3) remove the sample from the detection well , (4) move the wash buffer from wash buffer well to the detection well , (5) pipette up and down to wash the sensor (three times) and (6) move to the next sample. The SWV measurement was initiated at the Emstat interface (PSTrace) with a set time delay to synchronise with the robot. All the square-wave voltammograms obtained were normalised in PSTrace using the moving average baseline function and the peak value of each sample recorded. The peak values were processed and analysed via Excel, Origin and MedCal.

**Clinical samples**. Samples for data shown in Fig. 5 and Supplementary Figs. 4 and 5 were collected during 8-h research studies conducted in the clinical research facility at Imperial Healthcare NHS Trust, UK in accordance with Good Clinical Practice guidelines. Ethical approval was obtained prior, and all participants provided written informed consent before inclusion (Menopause study 15/LO/1481, ethical approval granted by Research Ethics Committee: London-West London and Gene Therapy Advisory Committee); healthy women with normal fertility 12/LO/0507, ethical approval granted by Research Ethics Committee: West London and women with hypothalamic amenorrhoea 13/LO/1807 and 12/LO/0507, ethical approval granted by Research Ethics Committee: London-West London and Gene Therapy Advisory Committee. Participants attended the clinical research facility and a peripheral cannula was inserted into a vein in their arm, which could then be used to repeatedly sample blood (every 10 min for 8 h = 49 samples/participant). The samples included had been taken from either the vehicle or baseline visit. Each blood sample (3 ml) taken for gonadotrophins (LH and FSH) and sex steroids (oestradiol and progesterone) was left to clot for at least 30 min, and then spun at 503 rcf for 10 min. The serum supernatant was then removed and immediately stored at −20 °C for subsequent analysis using a commercially available, automated chemiluminescent immunoassay method (Architect LH 2P40; Abbott Diagnostics, Maidenhead, UK). Laboratory analysis information was as follows: reference range: LH 4–14 IU/L, FSH 1.5–8 IU/L; intra-assay coefficient of variation: LH 4.1%, FSH 4.1%; inter-assay coefficient of variation LH 2.7%, FSH 3.0%; analytical sensitivity: LH 0.5 IU/L, FSH 0.05 IU/L. Architect LH Reagent Kit (2P40-25) Antibody details: LH 2P40, anti-β LH (mouse, monoclonal), minimum concentration 0.04% solids, anti-α LH (mouse, monoclonal), minimum concentration 170 ng/mL, specimen

dilution procedures: if required the automatic system performed a 1:4 dilution of the sample. Samples for data shown in Fig. 4 were archival serum from women undergoing fertility treatment ($n = 1$, six time points) and from women with hypothalamic amenorrhoea ($n = 3$) from the Hormone Laboratory, University Gynaecology Unit, Tsan Yuk Hospital, Hong Kong. Ethical approval for use and collection of these samples was obtained from the Institutional Review Board of the University of Hong Kong/Hospital Authority Hong Kong West Cluster (reference UW 15-652). These samples had been stored at −20 °C. The initial LH immunoassay had been performed by Architect LH Reagent Kit (2P40-25) on the automated chemiluminescent platform (Abbott Diagnostics, Illinois, USA).

**BSA**. A probability density distribution for inter-pulse intervals was obtained from both clinical assay and RAPTER assay datasets using BSA. We assume that the datagenerating process $d(t)$, can be written as the sum of an underlying oscillatory signal with frequency $\omega$, $s(t,\omega)$; a background drift function, $g(t)$ and a noise term, $e(t)$:

$$d(t) = s(t, \omega) + g(t) + e(t).$$

BSA calculates the posterior probability density distribution of $\omega$, for a given model using Bayes' rule:

$$P(\omega|D, M) = \frac{P(lD|\omega, M)P(\omega|M)}{P(D|M)},$$

where $P(D|\omega,M)$ is the probability of observing the data given model $M$, $P(\omega|M)$ is the prior distribution over frequencies (assumed uniform) and $P(D|M)$ is the evidence for model $M$. Furthermore, Bayesian model selection can be employed to choose the simplest, most constrained, model among models with different background functions. Analysis was performed using the BaSAR package in R[33], with sines and cosines as model functions, Legendre polynomials as background functions, and assuming time- independent Gaussian noise.

## Data availability

The authors declare that all data supporting the findings of this study are available within this published article, its Supplementary Information files and from the Source Data File. Supporting data are also available from the corresponding author upon reasonable request.

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

## Acknowledgements

This research was funded by Hong Kong Health and Medical Research Fund (HMRF) grant 03142546 to J.A.T. as a principal investigator. W.S.D. is funded by an NIHR Research Professorship (grant reference RP-2014-05-001) and the NIHR/Imperial Biomedical Research Centre. J.A.T. also acknowledges the University of Hong Kong Outstanding Young Researcher Award 2015-16 and HKU Seed Fund for Translational and Applied Research 201711160029. S.L. acknowledges the Imperial College London (ICL) and The University of Hong Kong (HKU) joint PhD programme funding support. K.T.A. and M.V. gratefully acknowledge the financial support of the EPSRC via grant EP/N014391/1. J.K.P. is funded by the MRC (grant reference MR/M024954/1) and an NIHR Academic Clinical Fellowship.

## Author contributions

S.L., A.E.G.C., W.S.D. and J.A.T. were responsible for the overall study design and data interpretation. S.L. and A.B.K. performed the SELEX and characterised the aptamers under supervision of J.A.T. S.L. and A.E.G.C. designed and fabricated the sensing platform. J.K.P. and W.S.D. collected clinical samples and were responsible for clinical aspects of the study in the United Kingdom. J.K.P. was responsible for the antibody comparison experiments under supervision of W.S.D. R.H.W.L. collected clinical samples and was responsible for clinical aspects of the study in Hong Kong. M.V., J.D.V., K.T.A. and C.A.M. developed the mathematical model and performed data analysis. A.E.G.C., W.S.D. and J.A.T. supervised the project. S.L., A.B.K., A.E.G.C., W.S.D. and J.A.T. drafted

the paper with critical feedback from all co-authors. All authors have critically read, edited and approved the final version of the paper.

## Additional information

**Competing interests:** S.L. is a co-founder of ZiO Health and J.A.T. is on the scientific advisory board of ZiO Health. W.S.D. is a consultant for Myovant Sciences Inc. and KaNDy Therapeutics Ltd. ZiO Health, Myovant Sciences and KaNDy Therapeutics were not involved in the research presented here, and the research area is in a topic unrelated to that of ZiO Health. The remaining authors declare no competing interests.

