## [Peer Review File · Nature Communications]

Reviewers' Comments:

Reviewer #1:

Remarks to the Author:

In this paper the authors have developed a new aptamer and its adaptation into an electrochemical aptasensor for the measurement of luteinising hormone, an important diagnostic biomarker. Their approach works in rather small samples, does not require any sample processing beyond conversion to serum, washing, or reagents, and appears easily parallelizable. It also achieves good limits of detection (relative to clinical need) and quantitation. They applied the technology to autonomously collected, minute-resolved human samples and demonstrate good correlation with gold standard approaches in a platform that is far easier and likely far less costly to use. Enough so that sufficiently high-frequency measurements can be made to easily determine luteinising hormone pulsatility.

This is solid work; it is rare to see biosensor papers for which the value-added proposition is so clear and for which authentic human samples have been measured and subjected to cross validation. That is, this approach is so much faster, cheaper, smaller-volume than existing methods that it represents a major advance, particularly for an application like this in which many dozens of samples must be measured in order to capture the relevant information (pulsatility). I thus strongly support the publication of this paper. I have only minor comments to suggest:

The authors claim an LOD of 10 nM. How is this defined?

Fig. 3, panel C the units are quoted as $-\%$, but since the values on the axis go negative, that's probably not true. In panel e in the caption they describe the units as $-\%$ but the units are μA in the actual figure.

Fig. 4 caption "the linear range is from 5 nM to 500 nM" but clearly it's not linear, it's hyperbolic. I.e., the physics themselves are not linear, and so quoting a linear range is only approximate and depends on how you define "linear" (ie how big a deviation you decide is significant). I'd recommend cutting that or at least saying something concrete.

Reviewer #2:

Remarks to the Author:

This is an outstanding paper by Liang and colleagues presenting data from an interdisciplinary team of internationally renowned experts, demonstrating the utility and efficacy of using aptamer enabled electrochemical sensing to measure clinical samples for LH pulsatility. They have not only developed and identified a novel and effective LH specific aptamer but engineered an electrochemical based sensor on a robotic platform, quite an achievement.

Additionally, they have successfully validated the system by comparing RAPTER measurements with the gold standard clinical immunochemical assays using rigorous Bayesian spectrum analysis. The quality and clarity of the data presented is outstanding, although colour contrast in some of the figures (eg. Fig 2b, fig 4c) is irritatingly poor.

The potential utility of this technology in clinical practice and its cost effectiveness is well articulated. The clinical ramifications for other hormone measurements and technical advancements make this paper particularly timely and of interest to a wide scientific and clinical audience in addition to interested lay public.

Minor comments:

1. Clarify that positive peak at 280nm increases after binding in figure 2d.
2. Briefly expand on how the technology could be adapted for continuous sensing. Do you mean via an implantable probe?
3. Parts of the methodology section could be refined considerably.
4. Polishing a 5mm length of wire with sandpaper sounds remarkable! What is the diameter of the

wire? The wire electrode is subsequently roughened with sulphuric acid, so why polish in the first place?

NCOMMS-18-36280-T

Response to Reviewers

We thank both reviewers for their excellent comments which help us to revise and improve the manuscript. We highlight their specific comments below with our response.

Reviewer #1:

Thank you for the positive comments overall.

The authors claim an LOD of 10 nM. How is this defined?

The LOD was calculated from 3 times standard deviation of the blank sample (PBS or HA serum without LH) / slope of the calibration line, we have indicated this in the revised manuscript.

Fig. 3, panel C the units are quoted as -%, but since the values on the axis go negative, that's probably not true. In panel e in the caption they describe the units as -% but the units are uA in the actual figure.

Thank you for pointing this out, we have corrected the figure caption.

Fig. 4 caption "the linear range is from 5 nM to 500 nM" but clearly it's not linear, it's hyperbolic. I.e., the physics themselves are not linear, and so quoting a linear range is only approximate and depends on how you define "linear" (ie how big a deviation you decide is significant). I'd recommend cutting that or at least saying something concrete.

We agree this is not linear and have cut as recommended.

Reviewer #2:

Thank you for the positive comments overall.

The quality and clarity of the data presented is outstanding, although colour contrast in some of the figures (eg. Fig 2b, fig 4c) is irritatingly poor.

We have improved the colour contrast.

Clarify that positive peak at 280nm increases after binding in figure 2d.

We corrected the typing mistake. The peak at 280 nm actually decreased after binding.

Briefly expand on how the technology could be adapted for continuous sensing. Do you mean via an implantable probe?

Yes, we have expanded on this in the discussion.

Parts of the methodology section could be refined considerably.

Methodology has been significantly edited throughout.

Polishing a 5mm length of wire with sandpaper sounds remarkable! What is the diameter of the wire? The wire electrode is subsequently roughened with sulphuric acid, so why polish in the first place?

The diameter is 0.5-1mm which we have indicated in manuscript. We have found the process of polishing followed by roughening effective and improves repeatability of the experiment.